# ENHANCING LoRA WITH SHARED RANDOM-SPAN AUGMENTATION FOR PARAMETER-EFFICIENT VISUAL ADAPTATION

## ABSTRACT

Low-rank Adaptation (LoRA) efficiently adapts large pre-trained models to down-stream tasks by learning low-rank adapters, significantly reducing computational and memory costs without sacrificing performance. Recent studies highlight the promise of rank adaptation methods in improving the flexibility and performance of LoRA. Grounded in Singular Value Decomposition (SVD) theory, these methods decompose the weight update into parameterized unitary matrices and learnable scaling coefficients, thereby allowing dynamic rank allocation of adapters based on coefficients. However, the parameterized construction of unitary matrices presents a significant computational bottleneck. To address this limitation, we propose Shared Random-Span Augmentation (SRSA), a novel Parameter-Efficient Fine-Tuning (PEFT) method that replaces the learnable unitary matrices with fixed, layer-shared random matrices. Our method facilitates flexible rank adaptation by learning scaling vectors within the shared random space, while maintaining parameter and memory efficiency. We provide both empirical and theoretical evidence to demonstrate the feasibility of substituting the unitary matrices with a shared random matrix. To evaluate the representational ability of our method, we conduct extensive experiments on various visual tasks. The results demonstrate that our method achieves compelling adaptation performance.

## 1 INTRODUCTION

The emergence of large-scale pre-trained vision models (Dosovitskiy et al., 2021; Liu et al., 2021; He et al., 2022) has significantly advanced the field of computer vision, while simultaneously creating a pressing need for parameter-efficient fine-tuning techniques (Houlsby et al., 2019; Chen et al., 2022; Hu et al., 2022; Liu et al., 2024). Such methods are essential to adapt these powerful models to diverse downstream tasks without incurring prohibitive computational costs. Among various efficient adaptation strategies, Low-Rank Adaptation (LoRA) (Hu et al., 2022) has gained considerable popularity due to its ability to dramatically reduce the number of trainable parameters while maintaining competitive performance. Despite its widespread adoption, a fundamental limitation of LoRA and similar fixed-rank approaches is their restricted capacity to generalize across tasks with varying complexities and data distributions. To elucidate this limitation, we analyze the orthogonality of the column vectors within LoRA's down-projection and up-projection matrices, building on insights from recent work (Yang et al., 2025). From Figure 1a, we observe that the angles between column vectors in LoRA adapters exhibit distinct distributions across different layers. As the dimension increases, the orthogonality progressively deteriorates. This phenomenon implies that the required rank varies across different adapter layers, demonstrating the inflexibility of the fixed-rank paradigm in adapting to diverse feature distributions. This drawback has motivated the development of more expressive fine-tuning mechanisms that can dynamically adjust their representational power.

Recent innovations (Zhang et al., 2023; Valipour et al., 2023; Dong et al., 2024a) have introduced the concept of learnable rank adaptation, which extends the conventional low-rank framework by allowing the singular values of the update matrices to be optimized during training. These methods leverage the principles of singular value decomposition to enable the rank of each adaptation matrix to vary based on task requirements. While this approach enhances flexibility, it introduces a significant computational burden: the orthogonal unitary matrices associated with the singular values must

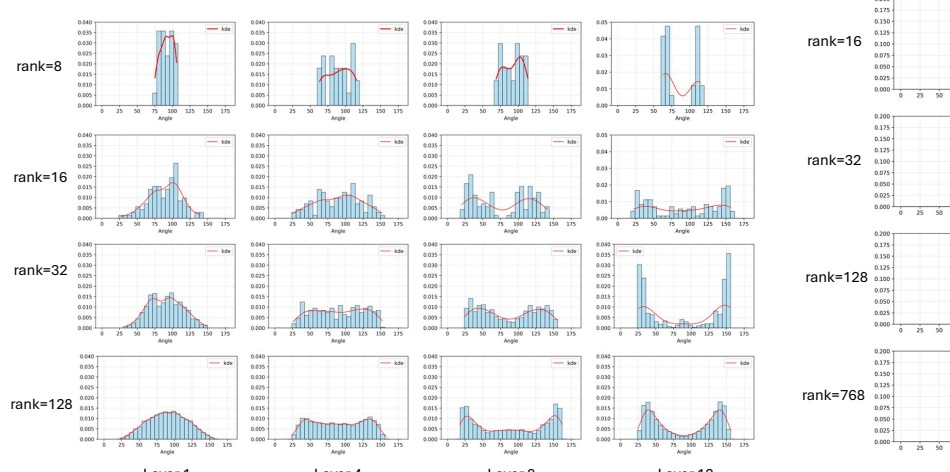

(a) The angle distribution across layers under different dimension configurations. The LoRA adapters are trained on KITTI dataset using ViT-B/16 pretrained backbone.

(b) The angle distribution of random initialized matrices under different dimension configurations.

Figure 1: Figure 1a shows the angle distribution between column vectors within the down/up-projection matrices in LoRA adapters. Figure 1b presents the angle distribution between column vectors of random initialized matrices.

also be updated, either explicitly or implicitly. This requirement not only increases the number of learnable parameters, but also complicates the optimization process, potentially leading to training instability and suboptimal convergence (He et al., 2025).

To address these challenges, we propose a novel fine-tuning framework termed Shared Random-Span Augmentation (SRSA). Our method offers a radical yet theoretically grounded alternative to conventional learnable rank mechanisms. Instead of optimizing the unitary matrices, SRSA replaces them with fixed, randomly initialized matrices that remain frozen throughout training. These random matrices are shared across different layers of the model, creating a highly efficient parameterization that minimizes introduced parameters. Crucially, we demonstrate that this design preserves the expressive power of adaptive-rank methods while substantially simplifying the learning process. We support our approach with a rigorous theoretical analysis that establishes the feasibility of using layer-shared random matrices as universal approximators in adaptive fine-tuning.

Through extensive experiments on a set of downstream vision classification tasks, we validate the effectiveness of SRSA. Our results show that the proposed method consistently outperform existing efficient adaptation approaches. The main contributions of this work are threefold:

1. We introduce SRSA, an efficient fine-tuning strategy that replaces learnable unitary bases with fixed, layer-shared random matrices, significantly reducing both computational and memory costs.

2. We provide a theoretical foundation for our approach, proving that frozen random matrices can effectively approximate the adaptive bases in learnable rank mechanisms without compromising representational power.

3. We conduct comprehensive experiments across multiple vision benchmarks, demonstrating that our SRSA can offer a compelling trade-off between parameter efficiency and representational capacity.

## 2 RELATED WORK

### 2.1 PARAMETER-EFFICIENT FINE-TUNING

The PEFT methods (Houlsby et al., 2019; Jia et al., 2022; Ben Zaken et al., 2022) effectively reduce resource overhead during downstream task adaptation by freezing the pre-trained weights and updat-

ing minimal introduced parameters. Among various PEFT techniques, Bias (Ben Zaken et al., 2022) focuses on fine-tuning only the bias terms of pre-trained model for specific downstream tasks, significantly reducing the training cost. VPT (Jia et al., 2022) introduces prompt learning into visual tasks. VPT achieves efficient adaptation through training task-specific prompts while keeping the model backbone frozen. SSF (Lian et al., 2022) accomplishes feature alignment with downstream tasks by performing scale and shift operations on the deep features of pre-trained model. ARC (Dong et al. (2023)) significantly reduces learnable parameter count through parameter sharing. Specifically, ARC shares unified trainable projection matrices across layers while employing tunable coefficients to capture layer-specific features. To further promote parameter efficiency, FACT (Jie & Deng, 2023) proposes a tensorization-decomposition framework, which factorizes the ViT weights into a 3D tensor while decomposing the weight increments into learnable lightweight factors. RLRR (Dong et al., 2024b) reveals the dynamics of PEFT methods from the perspective of SVD and proposes a Residual-based Low-Rank Rescaling (RLRR) fine-tuning strategy. AOFT (Yang et al., 2025) demonstrates that strict orthogonality whin adapter projection matrices can enhance model generalization. To this end, AOFT develops a efficient adaptation strategy which employs a single learnable vector to generate approximately orthogonal projection matrices.

## 2.2 LoRA AND LEARNABLE RANK ADAPTATION

LoRA (Hu et al., 2022) has gained significant popularity due to its competitive performance while maintaining parameter efficiency. In practice, LoRA represents original weights updates through the product of two learnable low-rank matrices, as illustrated in the Figure 2a. During the inference phase, these low-rank matrices are merged into the original weight matrices through reparameterization, thereby avoiding extra computational overhead. Building upon the success of LoRA (Hu et al., 2022), subsequent research has proposed various improvements to the original framework (Woo et al., 2025; Hayou et al., 2024). However, the fixed-rank paradigm of LoRA constrains its generalization capability when confronted with varying tasks and data distributions, which has motivated the development of various flexible rank adaptation strategies.

GoRA (He et al., 2025) leverages gradient information to statically assign ranks for low-rank adapters and initialize the corresponding weights. While GoRA's static rank adaptation mechanism introduces a degree of flexibility, it still fails to address the inherent limitations of such fixed-rank approaches. To overcome this challenge, DyLoRA (Valipour et al., 2023) and AdaLoRA (Zhang et al., 2023) employ learnable rank adaptation, allowing dynamic parameter budget allocation for adapters during training. HTA investigates the limitations of the fixed bottleneck dimension from the perspective of SVD. To address the limitation, HTA (Dong et al., 2024a) constructs Householder matrices using trainable vectors and adopts these matrices to efficiently mimic the unitary matrices. Additionally, HTA enables flexible rank adaptation through learnable diagonal vectors. While the aforementioned studies provide theoretical frameworks for flexible rank adaptation, these methods require introducing additional parameters to construct the high-dimensionality adaptive-space, which incurs additional computational overhead. In contrast, our work indicates that a random space is able to preserve the expressive power of adaptive-rank, offering a parameter-free construction paradigm.

## 2.3 PARAMETER-EFFICIENT FINE-TUNING USING RANDOM MATRICES

Random matrices are recognized as a promising technique in PEFT (Houlsby et al., 2019; Jia et al., 2022; Ben Zaken et al., 2022). Recent studies have replaced the learnable projection matrices with random initialized matrices that keep frozen during training, substantially reducing the number of trainable parameters. NoLA (Koohpayegani et al., 2024) reparametrizes projection matrices of LoRA (Hu et al., 2022) as linear combinations of a predefined set of random matrices. These random matrices remain frozen, while only the weighted coefficients for each matrix are learned during training. VERA (Kopiczko et al., 2024) freezes a single pair of low-rank random matrices that are shared across layers, while optimizing the scaling vectors that allow for layer-wise adaptation. RandLoRA (Albert et al., 2025) extends these approaches to achieve full-rank updates. Specifically, RandLoRA introduces a set of low-rank, non-trainable random matrices and learns scaling vectors to perform linear combinations of these matrices. In this work, we also present our SRSA fine-tuning strategy under the umbrella of random matrices. In contrast to previous attempts, we introduce a novel random matrices sharing strategy for learnable rank adaptation, along with theoretical insights

into this strategy. A visual comparison between our method and existing random matrix-based PEFT approaches is presented in Figure 2.

# 3 METHODOLOGY

## 3.1 PRELIMINARY

**LoRA.** For a weight matrix $\mathbf{W}_0 \in \mathbb{R}^{D_1 \times D_2}$ in a pre-trained Vision Transformer (ViT) model, fine-tuning optimizes $\mathbf{W}_0$ to obtain the weight update $\Delta \mathbf{W} \in \mathbb{R}^{D_1 \times D_2}$ tailored to the downstream task. LoRA freezes the pre-trained weights and approximates the weight update $\Delta \mathbf{W}$ using the product of two low-rank matrices, as follows:

$$\mathbf{W}' = \mathbf{W}_0 + \Delta \mathbf{W} \approx \mathbf{W}_0 + \mathbf{W}_{\text{down}} \mathbf{W}_{\text{up}}^\top, \tag{1}$$

where $\mathbf{W}'$ denotes the fine-tuned weight, $\mathbf{W}_{\text{down}} \in \mathbb{R}^{D_1 \times r}$ and $\mathbf{W}_{\text{up}} \in \mathbb{R}^{D_2 \times r}$ are low-rank matrices with $r \ll \min(D_1, D_2)$. As these low-rank matrices require fewer learnable parameters than original weights, LoRA drastically reduces the consumption of computational resources during fine-tuning.

**Singular Value Decomposition (SVD).** Given an arbitrary matrix $\mathbf{W} \in \mathbb{R}^{D_1 \times D_2}$, SVD algorithm factorizes it into a product of three matrices. Without loss of generality, we assume $D_1 < D_2$. Then SVD is represented as:

$$\mathbf{W} = \mathbf{U}\mathbf{S}\mathbf{V}^\top, \tag{2}$$

where $\mathbf{U} \in \mathbb{R}^{D_1 \times D_1}$ denotes the left unitary matrix, $\mathbf{V} \in \mathbb{R}^{D_2 \times D_2}$ represents the right unitary matrix, and $\mathbf{S} \in \mathbb{R}^{D_1 \times D_2}$ is a diagonal matrix with its diagonal entries arranged in descending order of singular values, such that $s_1 \geq s_2 \geq \cdots \geq s_{D_1} \geq 0$. Let $\mathbf{U} = [\boldsymbol{u}_1, \boldsymbol{u}_2, \ldots, \boldsymbol{u}_{D_1}]$ and $\mathbf{V} = [\boldsymbol{v}_1, \boldsymbol{v}_2, \ldots, \boldsymbol{v}_{D_2}]$, where $\boldsymbol{u}_i$ and $\boldsymbol{v}_i$ represent the bases vectors of $\mathbf{U}$ and $\mathbf{V}$, respectively. We can rewrite equation 2 as following:

$$\mathbf{W} = \sum_{i=1}^{D_1} \boldsymbol{u}_i s_i \boldsymbol{v}_i^\top. \tag{3}$$

## 3.2 LIMITATIONS OF LoRA

We assume the SVD form of the weight update $\Delta \mathbf{W}$ as $\sum_{i=1}^{D_1} \boldsymbol{u}_i s_i \boldsymbol{v}_i^\top$. To better reveal the limitations of LoRA, we present LoRA from the perspective of SVD, as expressed in :

$$\mathbf{W}_{\text{down}} \mathbf{W}_{\text{up}}^\top = \sum_{i=1}^{r} \boldsymbol{a}_i \boldsymbol{b}_i^\top, \tag{4}$$

where $\boldsymbol{a}_i$ and $\boldsymbol{b}_i$ are the column vectors of $\mathbf{W}_{\text{down}}$ and $\mathbf{W}_{\text{up}}$, respectively. The optimal approximation of LoRA to the weight update is achieved by minimizing the Frobenius norm of the difference between $\Delta \mathbf{W}$ and $\mathbf{W}_{\text{down}} \mathbf{W}_{\text{up}}^\top$:

$$\arg\min_{\mathbf{W}_{\text{down}}, \mathbf{W}_{\text{up}}} \|\Delta \mathbf{W} - \mathbf{W}_{\text{down}} \mathbf{W}_{\text{up}}^\top\|_F^2 = \arg\min_{\boldsymbol{a}_i, \boldsymbol{b}_i} \|\sum_{i=1}^{D_1} \boldsymbol{u}_i s_i \boldsymbol{v}_i^\top - \sum_{i=1}^{r} \boldsymbol{a}_i \boldsymbol{b}_i^\top\|_F^2. \tag{5}$$

According to the Eckart–Young–Mirsky theorem, the solution to equation 5 corresponds to the rank-$r$ truncation SVD of $\Delta \mathbf{W}$, i.e., $\sum_{i=1}^{r} \boldsymbol{a}_i \boldsymbol{b}_i^\top = \sum_{i=1}^{r} \boldsymbol{u}_i s_i \boldsymbol{v}_i^\top$, while the remaining components $\sum_{i=r+1}^{D_1} \boldsymbol{u}_i s_i \boldsymbol{v}_i^\top$ will be discarded. This finding indicates that LoRA captures the directional information of weight updates, which corresponds to the top-$k$ eigenvectors.

As observed, the expressive power of LoRA is constrained by assigned rank. The fixed-rank paradigm makes the adapter fails to approximate the shifting weight updates. Ideally, the rank of LoRA adapter should be dynamically adapted according to the effective rank of weight updates. However, this is impractical as the weight updates remain unknown before training. An alternative approach is to construct a high-dimensionality space and enable rank adaptation within this space to effectively model the dynamics in weight updates. This motivates us to investigate a feasible approach for efficiently building such a adaptive-space.

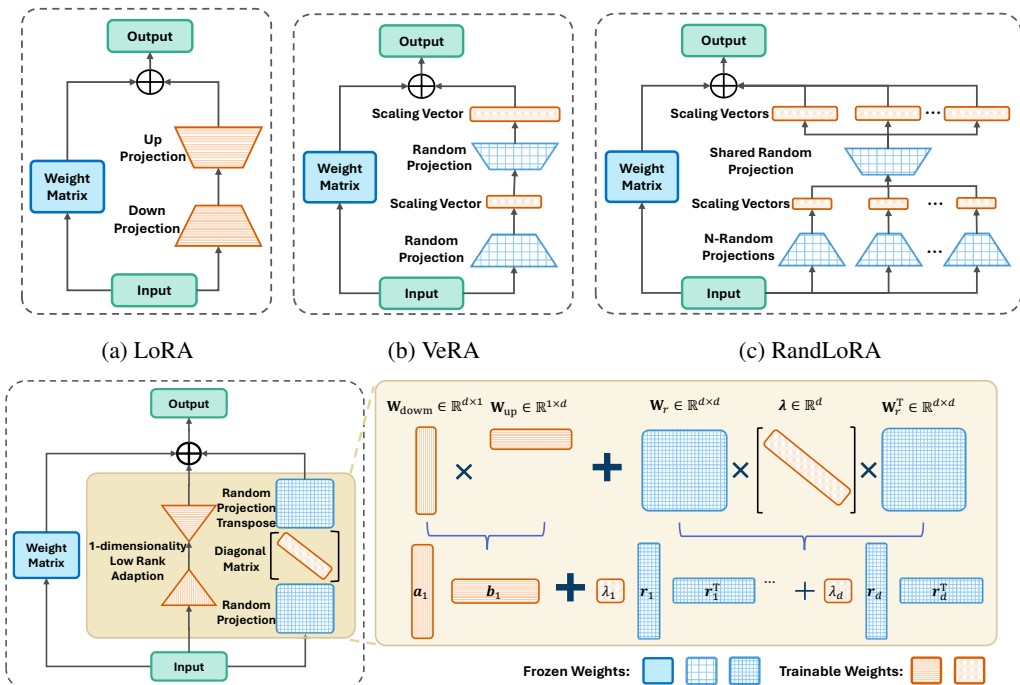

(a) LoRA       (b) VeRA       (c) RandLoRA

(d) Illustration of the proposed SRSA. Notably, the dimensionality of low-rank adaptation matrices can be flexibly configured for specific downstream tasks. We set the dimensionality to one as such setting achieves an effective balance between performance and parameter efficiency in the following experiments.

Figure 2: Visual comparison between our SRSA and existing PEFT methods.

## 3.3 ENHANCING LoRA WITH SHARED RANDOM-SPAN AUGMENTATION

Random matrices have demonstrated their ability to approximate adapter projection matrices in previous studies (Kopiczko et al., 2024; Albert et al., 2025). Furthermore, Figure 1b shows that the column vectors of random matrices preserve approximate orthogonality as the rank increases. This implies that a random initialized matrix with $d \times d$ size possesses the full-rank property. Based on this observation, we introduce frozen full-rank random matrices to replace the learnable unitary matrices in rank adaptation framework, which significantly reduces the computational overhead required for building the high-dimensional space. However, assigning layer-specific random matrices for each adapter layer incurs a non-negligible memory burden during fine-tuning. To this end, we propose an aggressive layer-wise sharing strategy: a unified random matrix is shared both across layers and within each layer. To demonstrate the feasibility of our sharing strategy, we provide a rigorous theoretical foundation bellow.

**Theorem 3.1.** Let $\mathbf{W}$ be an arbitrary matrix, $\mathbf{Q}$ an orthonormal matrix, and $\mathbf{E}$ a diagonal matrix. When both W and Q are fixed, approximating $\mathbf{W}$ with $\mathbf{QEQ}^\top$ can be defined as:

$$\arg\min_{\mathbf{E}} \|\mathbf{W} - \mathbf{QEQ}^\top\|_F^2. \tag{6}$$

The optimal solution to the equation 6 is given by $\mathbf{E} = \mathrm{diag}(\mathbf{QWQ}^\top)$, where the trace of $\mathbf{E}$ equals that of $\mathbf{W}$, i.e., $\mathrm{tr}(\mathbf{E}) = \mathrm{tr}(\mathbf{W})$. The detailed proof is provided in the Appendix C. This conclusion suggests that we can preserve the spectral property of the weight update $\Delta\mathbf{W}$, i.e., $\mathrm{tr}(\Delta\mathbf{W})$, by updating the diagonal matrix within rank adaptation framework. This property is independent of the unitary matrices, thereby providing the foundation for our proposed sharing strategy.

Inspired by these findings, we propose a novel PEFT approach that enhances the expressive power of LoRA with Shared Random-Span Augmentation (SRSA). The SRSA applies a rank adaptation mechanism to LoRA, as illustrated in Figure 2d. The mechanism comprises a left unitary matrix, a diagonal matrix, and a right unitary matrix. We replace the unitary matrices with non-trainable, layer-shared random matrices and update the diagonal matrix for efficient layer-wise adaptation.

Formally, our approach can be expressed as:

$$\text{SRSA} = \mathbf{W}_{\text{down}}\mathbf{W}_{\text{up}}^{\top} + \mathbf{W}_r\text{diag}(\boldsymbol{\lambda})\mathbf{W}_r^{\top} = \sum_{i=1}^{r} \boldsymbol{a}_i\boldsymbol{b}_i^{\top} + \sum_{j=1}^{d} \lambda_j \boldsymbol{r}_j\boldsymbol{r}_j^{\top}, \quad (7)$$

where $\mathbf{W}_r \in \mathbb{R}^{d \times d}$ denotes the shared random matrix, $\boldsymbol{\lambda} \in \mathbb{R}^d$ represents the learnable scaling vector, and $\boldsymbol{r}_j$ are the column vectors of the random matrix $\mathbf{W}_r$. During fine-tuning, we freeze the random matrix while updating the LoRA adapter and the learnable scaling vector. The SRSA enables LoRA adapter to capture the directional information of weight updates while simultaneously obtain the spectral propert through rank adaptation mechanism, thus enhancing the expressive power of LoRA.

## 4 EXPERIMENTS

### 4.1 EXPERIMENTAL SETTINGS

**Datasets.** We evaluate the effectiveness of our SRSA method on two sets of visual task adaptation benchmarks: FGVC and VTAB-1k, comprising 24 datasets in total. The FGVC benchmark comprises five Fine-Grained Visual Classification(FGVC) datasets, including CUB-200-2011(Wah et al., 2011), NABirds(Van Horn et al., 2015), Oxford Flowers(Nilsback & Zisserman, 2008), Stanford Dogs(Khosla et al., 2011), and Stanford Cars(Gebru et al., 2017). The VTAB-1k datasets(Zhai et al., 2019) consist of 19 visual classification tasks, divided into three groups: the Natural group, which contains natural images captured with standard cameras; the Specialized group, which includes images captured with specialized equipment; and the Structured group, which comprises synthesized images from simulated environments.

**Baselines and existing methods.**We conduct a comprehensive comparison of our SRSA method against two baselines and several state-of-the-art PEFT approaches. Two baselines we considered are: 1) Full fine-tuning, which updates all parameters in the pre-trained model; and 2) LoRA, which inserts learnable low-rank matrices into the adaptation layers while keeping the pre-trained weights frozen. In addition to baselines, we compare our method with the following state-of-the-art approaches: VPT(Jia et al., 2022), FacT(Jie & Deng, 2023), SSF, ARC(Dong et al., 2023), AOFT(Yang et al., 2025), RLRR(Dong et al., 2024b), HTA(Dong et al., 2024a).

**Our SRSA settings.** Depending on strategy of random matrix initialization, we present two variants of SRSA: SRSA-R and SRSA-SOR. For SRSA-R, we generate random matrices using Kaiming initialization, whose column vectors exhibit approximate orthogonality. Building upon SRSA-R, SRSA-SOR employs QR decomposition on the random matrices to obtain orthonormal matrices, which strictly satisfy the preconditions specified in **Theorem.3.1**. In addition, following previous work (Dong et al., 2024a), we set the dimension of LoRA adapter to one in our method and apply SRSZ ro the attention blocks.

More details on the dataset and experimental settings can be found in Appendix B.

### 4.2 EXPERIMENTAL COMPARISONS

In this section, we conduct extensive comparative experiments on image classification tasks. To evaluate the adaptation capacity of our SRSA method, we apply the SRSA strategy to two types of Vision Transformers: ViT (Dosovitskiy et al., 2021) and Swin Transformer (Liu et al., 2021) For ViT, we employ two backbones with different model sizes: ViT-B/16 and ViT-L/16, to demonstrate the generalization of our approach. All the backbones are pre-trained on the ImageNet-21K dataset.

**Comparisons with existing methods.** We evaluate the performance of our SRSA method against existing baselines and state-of-the-art solutions on the FGVC and VTAB-1k benchmarks. The results are presented in Table 1 and Table 2. We observe that SRSA demonstrates competitive classification accuracy with a reasonable parameter count. On the VTAB-1k collection, our SRSA achieves the best mean accuracy across all divided groups and the overall datasets. In particular, our method yields a significant accuracy improvement over the LoRA baseline, suggesting its effectiveness in enhancing the expressive power of LoRA. Compared to ARC, which also employs parameter-sharing techniques for efficient adaptation, the SRSA achieves a substantial performance

Table 1: Performance comparisons with state-of-the-art efficient adaptation methods on the VTAB-1k benchmark using the ViT-B/16 backbone pre-trained on ImageNet-21k. The **bold** font shows the best accuracy of all methods.

| Dataset / Method | Natural | | | | | | | | Specialized | | | | | Structured | | | | | | | | | Mean Total | Params.(M) |
|---|---|---|---|---|---|---|---|---|---|---|---|---|---|---|---|---|---|---|---|---|---|---|---|---|
| | CIFAR-100 | Caltech101 | DTD | Flowers102 | Pets | SVNH | Sun397 | Mean | Camelyon | EuroSAT | Resisc45 | Retinopathy | Mean | Clevr-Count | Clevr-Dist | DMLab | KITTI-Dist | dSpr-Loc | dSpr-Ori | sNORB-Azim | sNORB-Ele | Mean | | |
| Full fine-tuning | 68.9 | 87.7 | 64.3 | 97.2 | 86.9 | 87.4 | 38.8 | 75.9 | 79.7 | 95.7 | 84.2 | 73.9 | 83.4 | 56.3 | 58.6 | 41.7 | 65.5 | 57.5 | 46.7 | 25.7 | 29.1 | 47.6 | 65.6 | 85.8 |
| LoRA | 73 | 93.9 | 71.8 | 99.2 | 91.2 | 83.7 | 57.2 | 81.4 | 86 | 95.3 | 83.3 | 74.4 | 84.8 | 79.5 | 63.1 | 51.8 | 80.6 | 82.4 | 51.2 | 32.1 | 45 | 60.7 | 73.4 | 0.33 |
| Bias | 72.8 | 87.0 | 59.2 | 97.5 | 85.3 | 59.9 | 51.4 | 73.3 | 78.7 | 91.6 | 72.9 | 69.8 | 78.3 | 61.5 | 55.6 | 32.4 | 55.9 | 66.6 | 40.0 | 15.7 | 25.1 | 44.1 | 62.1 | 0.14 |
| VPT-Shallow | 77.7 | 86.9 | 62.6 | 97.5 | 87.3 | 74.5 | 51.2 | 76.8 | 78.2 | 92.0 | 75.6 | 72.9 | 79.7 | 50.5 | 58.6 | 40.5 | 67.1 | 68.7 | 36.1 | 20.2 | 34.1 | 47.0 | 64.9 | 0.11 |
| VPT-Deep | **78.8** | 90.8 | 65.8 | 98.0 | 88.3 | 78.1 | 49.6 | 78.5 | 81.8 | 96.1 | 83.4 | 68.4 | 82.4 | 68.5 | 60.0 | 46.5 | 72.8 | 73.6 | 47.9 | 32.9 | 37.8 | 55.0 | 69.4 | 0.60 |
| SSF | 58.0 | 89.8 | 70.5 | 98.9 | 90.2 | 90.5 | 52.9 | 78.7 | 86.7 | 95.2 | 86.4 | 75.4 | 85.9 | 68.2 | 61.0 | 52.8 | 80.7 | 77.3 | 48.5 | 27.6 | 31.1 | 55.9 | 70.6 | 0.24 |
| FacT-TK$_{\leq 32}$ | 70.6 | 90.6 | 70.8 | 99.1 | 90.7 | 88.6 | 54.1 | 80.6 | 84.8 | **96.2** | 84.5 | 75.7 | 85.3 | **82.6** | **68.2** | 49.8 | 80.7 | 80.8 | 47.4 | 33.2 | 43.0 | 60.7 | 73.2 | 0.07 |
| ARC | 72.2 | 90.1 | 72.7 | 99.0 | 91.0 | 91.9 | 54.4 | 81.6 | 84.9 | 95.7 | **86.7** | 75.8 | 85.8 | 80.7 | 67.1 | 48.7 | 81.6 | 79.2 | 51.0 | 31.4 | 39.9 | 60.0 | 73.4 | 0.13 |
| LoRA+AOFT | 74.2 | 93.4 | 72.7 | **99.4** | 91.5 | 85.5 | 57.3 | 82 | 86.3 | 95.2 | 84.0 | 75.8 | 85.3 | 78.9 | 63.1 | 51.2 | 82.6 | 83.6 | **53.9** | 31.9 | 47.3 | 61.6 | 74.1 | 0.08 |
| RLRR | 75.6 | 92.4 | 72.9 | 99.3 | 91.5 | 89.8 | 57.0 | 82.7 | 86.8 | 95.2 | 85.3 | 75.9 | 85.8 | 79.7 | 64.2 | **53.9** | 82.1 | 83.9 | 53.7 | 33.4 | 43.6 | 61.8 | 74.5 | 0.33 |
| HTA | 76.6 | 94.3 | 72.5 | 99.3 | 91.3 | 86.2 | 56.5 | 82.4 | 87.6 | 95.7 | 85.0 | 75.7 | 86.0 | **82.6** | 64.3 | 52.5 | 81.0 | 84.5 | 52.5 | 34.4 | 47.3 | 62.3 | 74.7 | 0.22 |
| SRSA-SOR | 76.7 | **94.7** | **73.4** | **99.4** | **91.9** | 86 | **57.6** | **82.8** | **87.9** | 95.5 | 85.9 | 76.1 | **86.4** | 82.5 | 64.2 | 51.8 | **82.7** | **84.9** | 52.7 | 34.2 | **48.3** | **62.7** | **75.1** | 0.15 |
| SRSA-R | 76.2 | 94.5 | 72.3 | 99.3 | 91.7 | 85.6 | 57.5 | 82.4 | 87.2 | 95.7 | 85.9 | **76.3** | 86.3 | 82.3 | 63.7 | 52.1 | 81.9 | 83.4 | 51.5 | **34.5** | **48.3** | 62.2 | 74.7 | 0.15 |

Table 2: Performance comparisons with state-of-the-art efficient adaptation methods on the FGVC datasets using the ViT-B/16 backbone pre-trained on ImageNet-21k.

| Dataset / Method | CUB-200-2011 | NABirds | Oxford Flowers | Stanford Dogs | Stanford Cars | Mean | Params.(M) |
|---|---|---|---|---|---|---|---|
| Full fine-tuning | 87.3 | 82.7 | 98.8 | 89.4 | 84.5 | 88.5 | 85.98 |
| LoRA | 88.3 | 85.6 | 99.2 | 91.0 | 83.2 | 89.5 | 0.44 |
| Bias | 88.4 | 84.2 | 98.8 | 91.2 | 79.4 | 88.4 | 0.28 |
| VPT-Shallow | 86.7 | 78.8 | 98.4 | 90.7 | 68.7 | 84.6 | 0.25 |
| VPT-Deep | 88.5 | 84.2 | 99.0 | 90.2 | 83.6 | 89.1 | 0.85 |
| SSF | 82.7 | **85.9** | 98.5 | 87.7 | 82.6 | 87.5 | 0.39 |
| LoRA+AOFT | 88.8 | 84.2 | 99.4 | 92.0 | 85.1 | 89.9 | 0.22 |
| ARC | 88.5 | 85.3 | 99.3 | 91.9 | 85.7 | 90.1 | 0.20 |
| RLRR | **89.3** | 84.7 | **99.5** | 92.0 | 87.0 | 90.4 | 0.47 |
| HTA | 88.8 | 84.4 | **99.5** | 92.2 | 87.9 | 90.6 | 0.36 |
| SRSA-SOR | 89.1 | 84.7 | **99.5** | **92.5** | **88.1** | **90.8** | 0.29 |
| SRSA-R | 88.9 | 84.9 | **99.5** | 92 | 87.7 | 90.6 | 0.29 |

gain while maintaining a comparable parameter budget. Furthermore, the proposed SRSA outperforms previous state-of-the-art work in general, such as AOFT, RLRR, etc. The results summarized in Table 2 indicate that our method still delivers appealing performance within the context of performance saturation observed on the FGVC benchmark. Notably, the SRSA-SOR variant exhibits superior performance compared to the SRSA-R variant. We attribute this improvement to the fact that orthonormal random matrices employed in STSA-SOR rigorously satisfy the assumptions presented in **Theorem.3.1**.

**Experiments on larger-scale ViT backbone.** To validate the adaptation capacity of our approach across different scales of pre-trained models, we also conduct experiments using the ViT-L/16 backbone. A summary of results is available in Table 3 with detailed results being available in Appendix B. Experimental results show that our method consistently outperforms existing approaches while maintaining parameter efficiency. This observation suggests that our SRSA exhibits robust expressive capabilities even when scaled to larger backbones.

**Experiments on hierarchical Vision Transformers.** To evaluate the generality of our approach across different network architectures, we conducted comparative experiments on Swin Transformer, a hierarchical Transformer architecture. To accommodate the varying feature dimensionalities in Swin Transformer, we introduced a stage-sharing strategy, enabling random matrices sharing within each stage. As shown in Table 4, the SRSA-SOR achieves state-of-the-art performance on the VTAB-1k benchmark and maintains favorable parameter scale. Compared to HTA, our method yields a 0.8% improvement in average accuracy. Experimental results highlight the versatility of our SRSA in adapting different transformer architectures, suggesting its practical potential for visual adaptation tasks.

Table 3: Performance comparison on VTAB-1k using ViT-L/16 pre-trained on ImageNet-21k as the backbone. "(·)" denotes the number of tasks in the subgroup.

| | Natural (7) | Specialized (4) | Structed (8) | Mean Total | Params. |
|---|---|---|---|---|---|
| Full fine-tuning | 74.7 | 83.8 | 48.1 | 65.4 | 303.40 |
| LoRA | 81.4 | 85.0 | 57.3 | 72.0 | 0.74 |
| Bias | 70.5 | 73.8 | 41.2 | 58.9 | 0.32 |
| VPT-Shallow | 78.7 | 79.9 | 40.6 | 62.9 | 0.15 |
| VPT-Deep | 82.5 | 83.9 | 54.1 | 70.8 | 0.49 |
| SSF | 81.9 | 85.2 | 59.0 | 73.0 | 0.60 |
| ARC | 82.3 | 85.6 | 57.3 | 72.5 | 0.18 |
| LoRA+AOFT | 83.3 | 85.9 | 60.2 | 74.3 | 0.15 |
| RLRR | 83.9 | 86.4 | 61.9 | 75.2 | 0.82 |
| HTA | 84.1 | 86.6 | 62.3 | 75.4 | 0.54 |
| SRSA-SOR | **84.5** | **86.8** | **63.1** | **76** | 0.30 |

Table 4: Performance comparison on VTAB-1k using Swin-B pre-trained on ImageNet-21k as the backbone.

| | Natural (7) | Specialized (4) | Structed (8) | Mean Total | Params. |
|---|---|---|---|---|---|
| Full fine-tuning | 79.1 | 86.2 | 59.7 | 72.4 | 86.80 |
| LoRA | 81 | 85.2 | 60.5 | 73.2 | 0.82 |
| Bias | 74.2 | 80.1 | 42.4 | 62.1 | 0.25 |
| VPT-Shallow | 79.9 | 82.5 | 37.8 | 62.9 | 0.05 |
| VPT-Deep | 76.8 | 84.5 | 53.4 | 67.7 | 0.22 |
| ARC | 79.0 | 86.6 | 59.9 | 72.6 | 0.27 |
| LoRA+AOFT | 82.3 | 86.8 | 60.6 | 73.3 | 0.14 |
| RLRR | 81.3 | 86.7 | 59.0 | 73.0 | 0.41 |
| HTA | 81.8 | 86.7 | 61.3 | 74.2 | 0.23 |
| SRSA-SOR | **82.6** | **86.9** | **62.4** | **75** | 0.19 |

## 4.3 ABLATION STUDIES

**Effect of random matrices sharing strategy.** In proposed SRSA, we introduce a layer-wise sharing strategy for random matrices and provide theoretical analysis for its feasibility in rank adaptation. The layer-wise sharing strategy encompasses two aspects: intra-layer sharing, where the left and right unitary matrices within one adapter are transposes of each other; and inter-layer sharing, where different adapter layers share the unified random matrix. We conducted a systematic comparison of different sharing schemes to further investigate the rationale behind the sharing strategy, as detailed in Table 5. The results indicate that disabling either intra-layer or inter-layer sharing does not result in performance gains. In contrast, introducing unshared random matrices will increase the training overhead. This validates the effectiveness of our random matrices sharing strategy.

**Effect of adapter positioning.** As a plug-and-play method, our SRSA offers flexibility akin to LoRA, enabling seamless integration into different model components. We investigate the performance of our SRSA when placed in various components. For comparison, we include standard LoRA applied to $\{\mathbf{W}_q, \mathbf{W}_v\}$ projection matrices in the multi-head attention operation of each ViT layer as the baseline. As shown in Table 6, under same integration components with the baseline, our method demonstrates superior performance to LoRA while using fewer learnable parameters. Furthermore, when we extend SRSA adapters to $\{\mathbf{W}_k, \mathbf{W}_o\}$ and $\{\mathbf{W}_{FC_1}, \mathbf{W}_{FC_2}\}$, results exhibit a substantial performance improvement. These findings suggest that our method allows a better trade-off between adaptation performance and parameter efficiency by flexibly selecting the integration components.

**Effect of bottleneck dimensionality.** To investigate the impact of LoRA's dimensionality on our SRSA method, we systematically examine model performance across different bottleneck dimen-

Table 5: Ablation study on random matrices sharing strategies. All experiments are conducted on VTAB-1k benchmark using ViT-B/16 backbone.

| | Natural (7) | Specialized (4) | Structed (8) | Mean Total | Params. |
|---|---|---|---|---|---|
| SRSA-SOR | **82.6** | **86.8** | **62.7** | **75.1** | 0.15 |
| w/o intra + inter | 82.2 | 85.8 | 61.9 | 74.4 | 0.15 |
| intra + w/o inter | 82.4 | 85.9 | 62.2 | 74.6 | 0.15 |

Table 6: Ablation Study on adapter positioning. All experiments are conducted on VTAB-1k benchmark using ViT-B/16 backbone.

| | Natural (7) | Specialized (4) | Structed (8) | Mean Total | Params. |
|---|---|---|---|---|---|
| LoRA($\mathbf{W}_q, \mathbf{W}_v$) | 79.5 | 84.6 | 59.8 | 72.3 | 0.29 |
| SRSA-SOR($\mathbf{W}_q, \mathbf{W}_v$) | 81.4 | 85.3 | 59.7 | 73.1 | 0.08 |
| SRSA-SOR($\mathbf{W}_q, \mathbf{W}_v, \mathbf{W}_{FC_1}, \mathbf{W}_{FC_2}$) | 82.1 | 86.3 | 61.0 | 74.1 | 0.20 |
| SRSA-SOR($\mathbf{W}_q, \mathbf{W}_k, \mathbf{W}_v, \mathbf{W}_o$) | 82.8 | 86.4 | 62.7 | 75.1 | 0.15 |

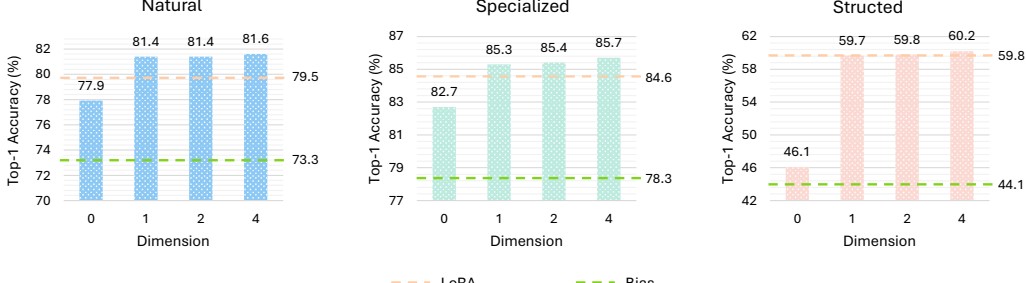

Figure 3: Ablation study on the impact of different bottleneck dimensions of LoRA adapter. The bar chart represents the Top-1 Test Accuracy. We apply SRSA to the $\{\mathbf{W}_q, \mathbf{W}_v\}$ matrices.

sions, as shown in the Figure 3. For a fair comparison, we apply our SRSA to the $\{\mathbf{W}_q, \mathbf{W}_v\}$ matrices. Additionally, we include a comparison with the Bias (Ben Zaken et al., 2022), a PEFT method that updates only the bias terms of pre-trained model. We observe a significant performance drop in SRSA when we disable LoRA by setting its dimensionality to zero. This finding corroborates our claim in Section 3.3 that while SRSA excels at rank adaptation, it relies on LoRA to provide the crucial directional updates. Remarkably, our SRSA achieves a significant performance gain over the Bias method, even with only the rank adaptation component active. When the dimension increases to one, our SRSA performance rises significantly, surpassing the LoRA baseline. However, as the dimension increases further, the performance gains become marginal. We attribute this phenomenon to the performance bottleneck in LoRA.

## 5 LIMITATIONS

Our method employs a novel layer-wise sharing strategy that shares a unique random matrix both across layers and within layers. However, the sharing scheme is built on the assumption that layers have the same dimensionality. For hierarchical model architectures where feature dimensions vary across layers, it is necessary to explore efficient alternatives to the sharing strategy. Additionally, although our method enhances the expressiveness of LoRA through rank adaptation mechanism, our approach critically depends on LoRA to learn the directional information of weight updates. This implies that the performance bottleneck of LoRA limits the upper bound of our model's generalization capability. To address the limitation, it is worth exploring strategies to integrate our SRSA into advanced LoRA variants to further improve the model performance.

## 6 CONCLUSION

In this work, we propose a novel fine-tuning method called Shared Random-Span Augmentation (SRSA). Our approach is specifically designed to overcome the computational bottleneck posed by learnable unitary matrices in existing rank adaptation frameworks. Specifically, our method employs fixed random matrices as a substitute for learnable unitary matrices and shares these matrices both across and within layers, thereby significantly reducing the number of learnable parameters. We provide empirical evidence to demonstrate that fixed random matrices can effectively approximate learnable unitary structures, along with theoretical justification for the feasibility of sharing random matrices across layers. Experimental results show that our SRSA achieves state-of-the-art performance across various visual downstream tasks.

## 7 REPRODUCIBILITY STATEMENT

To facilitate reproducibility of our method, we submit our source code to supplementary material in a compressed file for the double-blind review and will open-source it after paper publication. The compressed file contains the code for data processing, model implementation, and essential evaluation. We recommend downloading the corresponding pre-trained models and datasets from open-source repositories in (Dong et al., 2024b; 2023). All implementation details of our work are meticulously described in Section B.

## 8 ETHICS STATEMENT

We confirm that all authors have read and adhered to the ICLR Code of Ethics and explicitly complied with its requirements throughout all conference activities (including submission, review, and discussion). All datasets in this study were sourced from publicly available repositories under licenses permitting research use. Therefore, no specific ethical approval was required for data acquisition and analysis.

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

## A USE OF LARGE LANGUAGE MODELS

In this paper, Large Language Models (e.g., DeepSeek, ChatGPT) were used exclusively as tools for polishing the writing. The literature findings and the core ideas of this paper were entirely conceived and developed by the authors.

Table 7: Dataset statistics for FGVC. "*" denotes the train/val split of datasets following the dataset setting of VPT models Jia et al. (2022).

| Dataset | Description | Classes | Train size | Val size | Test size |
|---|---|---|---|---|---|
| CUB-200-2011 | Fine-grained bird species recognition | 200 | 5,394* | 600* | 5,794 |
| NABirds | Fine-grained bird species recognition | 555 | 21,536* | 2,393* | 24,633 |
| Oxford Flowers | Fine-grained flower species recognition | 102 | 1,020 | 1,020 | 6,149 |
| Stanford Dogs | Fine-grained dog species recognition | 120 | 10,800* | 1,200* | 8,580 |
| Stanford Cars | Fine-grained car classificatio | 196 | 7,329* | 815* | 8,041 |

Table 8: Dataset statistics for VTAB-1k Zhai et al. (2019).

| Dataset | Description | Classes | Train size | Val size | Test size |
|---|---|---|---|---|---|
| CIFAR-100 | | 100 | | | 10,000 |
| Caltech101 | | 102 | | | 6,084 |
| DTD | | 47 | | | 1,880 |
| Flowers102 | Natural | 102 | 800/1,000 | 200 | 6,149 |
| Pets | | 37 | | | 3,669 |
| SVHN | | 10 | | | 26,032 |
| Sun397 | | 397 | | | 21,750 |
| Patch Camelyon | | 2 | | | 32,768 |
| EuroSAT | Specialized | 10 | 800/1,000 | 200 | 5,400 |
| Resisc45 | | 45 | | | 6,300 |
| Retinopathy | | 5 | | | 42,670 |
| Clevr/count | | 8 | | | 15,000 |
| Clevr/distance | | 6 | | | 15,000 |
| DMLab | | 6 | | | 22,735 |
| KITTI/distance | | 4 | | | 711 |
| dSprites/location | Structured | 16 | 800/1,000 | 200 | 73,728 |
| dSprites/orientation | | 16 | | | 73,728 |
| SmallNORB/azimuth | | 18 | | | 12,150 |
| SmallNORB/elevation | | 9 | | | 12,150 |

## B    Details of experimental settings

### B.1    Details of datasets

We delineate the experimental specifications for visual adaptation classification tasks in Table 7 (FGVC) and Table 8 (VTAB-1k), including the cardinalities of the categories and training / validation / test set partitions, establishing standardized benchmarks for reproducible parameter-efficient fine-tuning evaluation. The FGVC benchmark suite comprises five fine-grained visual recognition datasets: CUB-200-2011, NABirds, Oxford Flowers, Stanford Dogs, and Stanford Cars, each targeting specialized classification tasks for avian species, floral categories, canine breeds, and automotive models respectively. In parallel, the VTAB-1k framework organizes its evaluation protocol into three task domains:Natural Image Domain: CIFAR-100, Caltech101, DTD, Flowers102, Pets, SVHN, and Sun397; Specialized Image Domain: Patch Camelyon, EuroSAT, Resisc45, and Retinopathy;Structured Image Domain: Clevr / count, Clevr / distance, DMLab, KITTI / distance, dSprites / location, dSprites / orientation, SmallNORB / azimuth and SmallNORB / elevation.

### B.2    Implementation details

To facilitate an impartial evaluation of our proposed SRSA, we follow previous work (Dong et al., 2023; 2024b;a) and employ simple data augmentation during the data processing stage. For the FGVC datasets, we processed the images with a random resize crop to $224 \times 224$ and applied a random horizontal flip. For the VTAB-1k datasets, we directly resized the images to $224 \times 224$. To ensure consistency with prior work, we employ a grid search to identify the optimal hyperparameter settings, including batch size, learning rate, dropout rate and weight decay. The details of hyperparameter settings are shown in Tabel 9. All experiments in our work are carried out on NVIDIA 4090 GPUs (24GB VRAM) using PyTorch 3.8.

Table 9: The implementation details of configurations such as optimizer and hyperparameter. We select the best hyperparameter settings for each download task by using grid search.

| | |
|---|---|
| Optimizer | AdamW |
| Learning Rate | {0.01, 0.005, 0.003, 0.001, 0.0005, 0.0003, 0.0001} |
| Weight Decay | {0.05, 0.01, 0.005, 0.001, 0} |
| Batch Size | {64, 32, 16} |
| Dropout rate | {0.5, 0.4, 0.3, 0.2, 0.1, 0} |
| Learning Rate Schedule | Cosine Decay |
| Training Epochs | 100 |
| Warmup Epochs | 10 |

## C    Mathematical derivations and proofs

### C.1    Theorem 3.1

In this section, we give the detailed proof of **Theorem 3.1** in 3.3 of the main paper.

**Theorem 3.1.** Let $\mathbf{W}$ be an arbitrary matrix, $\mathbf{Q}$ an orthonormal matrix, and $\mathbf{E}$ a diagonal matrix. When both W and Q are fixed, approximating $\mathbf{W}$ with $\mathbf{QEQ}^{\top}$ can be defined as:

$$\arg\min_{\mathbf{E}} \|\mathbf{W} - \mathbf{QEQ}^{\top}\|_F^2. \tag{8}$$

The optimal solution to the equation 6 is given by $\mathbf{E} = \text{diag}(\mathbf{QWQ}^{\top})$, where the trace of $\mathbf{E}$ equals that of $\mathbf{W}$, i.e., $\text{tr}(\mathbf{E}) = \text{tr}(\mathbf{W})$.

*Proof.* According to the definition of the matrix Frobenius norm, we have $\|\mathbf{W}\|_F^2 = \text{tr}(\mathbf{W}^{\top}\mathbf{W})$. Consequently, Equation 8 can be derived as follows:

$$\begin{aligned}
\|\mathbf{W} - \mathbf{QEQ}^{\top}\|_F^2 &= \text{tr}((\mathbf{W} - \mathbf{QEQ}^{\top})^{\top}(\mathbf{W} - \mathbf{QEQ}^{\top})) \\
&= \text{tr}(\mathbf{W}^{\top}\mathbf{W} - \mathbf{W}^{\top}\mathbf{QEQ}^{\top} - (\mathbf{QEQ}^{\top})^{\top}\mathbf{W} + (\mathbf{QEQ}^{\top})^{\top}\mathbf{QEQ}^{\top}).
\end{aligned} \tag{9}$$

Table 10: This table is extended from Table 3 and shows performance comparisons on the VTAB-1k benchmark using the ViT-L/16 backbone pre-trained on ImageNet-21k.

| | Natural | | | | | | | | Specialized | | | | | Structured | | | | | | | | | Mean Total | Params.(M) |
|---|---|---|---|---|---|---|---|---|---|---|---|---|---|---|---|---|---|---|---|---|---|---|---|---|
| Dataset / Method | CIFAR-100 | Caltech101 | DTD | Flowers102 | Pets | SVNH | Sun397 | Mean | Camelyon | EuroSAT | Resisc45 | Retinopathy | Mean | Clevr-Count | Clevr-Dist | DMLab | KITTI-Dist | dSpr-Loc | dSpr-Ori | sNORB-Azim | sNORB-Ele | Mean | | |
| Full fine-tuning | 68.6 | 84.3 | 58.6 | 96.3 | 86.5 | 87.5 | 41.4 | 74.7 | 82.6 | 95.9 | 82.4 | 74.2 | 83.8 | 55.4 | 55.0 | 42.2 | 74.2 | 56.8 | 43.0 | 28.5 | 29.7 | 48.1 | 65.4 | 303.4 |
| LoRA | 75.8 | 89.8 | 73.6 | 99.1 | 90.8 | 83.2 | 57.5 | 81.4 | 86.0 | 95.0 | 83.4 | 75.5 | 85.0 | 78.1 | 60.5 | 46.7 | 81.6 | 76.7 | 51.3 | 28.0 | 35.4 | 57.3 | 72.0 | 0.74 |
| Bias | 71.0 | 82.4 | 51.3 | 96.3 | 83.2 | 59.5 | 49.9 | 70.5 | 72.9 | 87.9 | 63.1 | 71.3 | 73.8 | 51.2 | 50.7 | 33.5 | 54.8 | 65.9 | 37.3 | 13.7 | 22.2 | 41.2 | 58.9 | 0.32 |
| VPT-Shallow | 80.6 | 88.2 | 67.1 | 98.0 | 85.9 | 78.4 | 53.0 | 78.7 | 79.7 | 93.5 | 73.4 | 73.1 | 79.9 | 41.5 | 52.5 | 32.3 | 64.2 | 48.3 | 35.3 | 21.6 | 28.8 | 40.6 | 62.9 | 0.15 |
| VPT-Deep | 84.1 | 88.9 | 70.8 | 98.8 | 90.0 | 89.0 | 55.9 | 82.5 | 82.5 | 96.6 | 82.6 | 73.9 | 83.9 | 63.7 | 60.7 | 46.1 | 75.7 | 83.7 | 47.4 | 18.9 | 36.9 | 54.1 | 70.8 | 0.49 |
| ARC | 76.2 | 89.6 | 73.4 | 99.1 | 90.3 | 90.9 | 56.5 | 82.3 | 85.0 | 95.7 | 85.9 | 75.8 | 85.6 | 78.6 | 62.1 | 46.7 | 76.7 | 75.9 | 53.0 | 30.2 | 35.2 | 57.3 | 72.5 | 0.18 |
| LoRA+AOFT | 78.2 | 95.0 | 74.7 | 99.5 | 92.0 | 82.4 | 59.2 | 83.3 | 86.7 | 95.1 | 86.0 | 75.2 | 85.9 | 81.5 | 63.2 | 50.7 | 81.0 | 86.7 | 53.0 | 28.8 | 43.3 | 60.2 | 74.3 | 0.15 |
| RLRR | 79.3 | 92.0 | 74.6 | 99.5 | 92.1 | 89.6 | 60.1 | 83.9 | 87.3 | 95.3 | 87.3 | 75.7 | 86.4 | 82.7 | 62.1 | 54.6 | 80.6 | 87.1 | 54.7 | 31.3 | 41.9 | 61.9 | 75.2 | 0.82 |
| HTA | 80.8 | 92.4 | 76.1 | 99.5 | 92.8 | 87.2 | 59.9 | 84.1 | 87.7 | 95.5 | 86.8 | 76.5 | 86.6 | 82.6 | 62.4 | 53.4 | 80.0 | 87.1 | 53.7 | 33.4 | 45.6 | 62.3 | 75.4 | 0.54 |
| SRSA-SOR | 79.8 | 95,2 | 76.0 | 99.5 | 92.9 | 87.6 | 60.2 | 84.5 | 87.6 | 95.7 | 87.0 | 76.9 | 86.8 | 83.5 | 63.4 | 53.6 | 82.3 | 87.1 | 53.5 | 35.1 | 46.2 | 63.1 | 76.0 | 0.30 |

Table 11: This table is extended from Table 4 and shows performance comparisons on the VTAB-1k benchmark using the Swin-B backbone pre-trained on ImageNet-21k.

| | Natural | | | | | | | | Specialized | | | | | Structured | | | | | | | | | Mean Total | Params.(M) |
|---|---|---|---|---|---|---|---|---|---|---|---|---|---|---|---|---|---|---|---|---|---|---|---|---|
| Dataset / Method | CIFAR-100 | Caltech101 | DTD | Flowers102 | Pets | SVNH | Sun397 | Mean | Camelyon | EuroSAT | Resisc45 | Retinopathy | Mean | Clevr-Count | Clevr-Dist | DMLab | KITTI-Dist | dSpr-Loc | dSpr-Ori | sNORB-Azim | sNORB-Ele | Mean | | |
| Full fine-tuning | 72.2 | 88.0 | 71.4 | 98.3 | 89.5 | 89.4 | 45.1 | 79.1 | 86.6 | 96.9 | 87.7 | 73.6 | 86.2 | 75.7 | 59.8 | 54.6 | 78.6 | 79.4 | 53.6 | 34.6 | 40.9 | 59.7 | 72.4 | 86.9 |
| LoRA | 70.1 | 89.9 | 73.8 | 99.3 | 91.3 | 87.4 | 54.9 | 81.0 | 85.8 | 95.0 | 84.3 | 75.5 | 85.2 | 83.2 | 60.1 | 53.1 | 83.1 | 89.5 | 50.8 | 26.9 | 37.2 | 60.5 | 73.2 | 0.82 |
| Bias | 73.1 | 86.8 | 65.7 | 97.7 | 87.5 | 56.4 | 52.3 | 74.2 | 80.4 | 91.6 | 76.1 | 72.5 | 80.1 | 47.3 | 48.5 | 34.7 | 66.3 | 57.6 | 36.2 | 17.2 | 31.6 | 42.4 | 62.1 | 0.25 |
| VPT-Shallow | 78.0 | 91.3 | 77.2 | 99.4 | 90.4 | 68.4 | 54.3 | 79.9 | 80.1 | 93.9 | 83.0 | 72.7 | 82.5 | 40.8 | 43.9 | 34.1 | 63.2 | 28.4 | 44.5 | 21.5 | 26.3 | 37.8 | 62.9 | 0.05 |
| VPT-Deep | 79.6 | 90.8 | 78.0 | 99.5 | 91.4 | 46.5 | 51.7 | 76.8 | 84.9 | 96.2 | 85.0 | 72.0 | 84.5 | 67.6 | 59.4 | 50.1 | 74.1 | 74.4 | 50.6 | 25.7 | 25.7 | 53.4 | 67.7 | 0.22 |
| ARC | 62.5 | 90.0 | 71.9 | 99.2 | 87.8 | 90.7 | 51.1 | 79.0 | 89.1 | 95.8 | 84.5 | 77.0 | 86.6 | 75.4 | 57.4 | 53.4 | 83.1 | 91.7 | 55.2 | 31.6 | 31.8 | 59.9 | 72.6 | 0.27 |
| LoRA+AOFT | 71.8 | 92.3 | 77.1 | 99.5 | 92.6 | 86.4 | 55.8 | 82.3 | 86.9 | 96.4 | 87.3 | 77.6 | 86.8 | 84.5 | 59.3 | 53.6 | 84.7 | 86.8 | 52.3 | 28.1 | 35.5 | 60.6 | 73.3 | 0.14 |
| RLRR | 66.1 | 90.6 | 75.5 | 99.3 | 92.1 | 90.9 | 54.7 | 81.3 | 87.1 | 95.9 | 87.1 | 76.5 | 86.7 | 66.0 | 57.8 | 55.3 | 84.1 | 91.1 | 55.2 | 28.6 | 34.0 | 59.0 | 73.0 | 0.41 |
| HTA | 72.0 | 89.6 | 76.4 | 99.5 | 92.1 | 87.8 | 55.5 | 81.8 | 86.7 | 96.3 | 87.5 | 76.3 | 86.7 | 85.0 | 62.2 | 53.7 | 84.3 | 89.1 | 52.4 | 27.6 | 36.4 | 61.3 | 74.2 | 0.23 |
| SRSA-SOR | 73.2 | 91 | 76.7 | 99.5 | 92.4 | 89.3 | 55.8 | 82.6 | 87.3 | 96.1 | 87.8 | 76.3 | 86.9 | 85.8 | 62.9 | 53.9 | 84.5 | 92.5 | 53.3 | 28.6 | 37.9 | 62.4 | 75.0 | 0.19 |

Based on the properties of the matrix trace, we obtain $\mathrm{tr}(\mathbf{W}^\top) = \mathrm{tr}(\mathbf{W})$ and $\mathrm{tr}(\mathbf{AB}) = \mathrm{tr}(\mathbf{BA})$. Consequently, Equation 9 can be derived as follows:

$$\|\mathbf{W} - \mathbf{QEQ}^\top\|_F^2 = \mathrm{tr}(\mathbf{W}^\top \mathbf{W}) - 2\mathrm{tr}((\mathbf{QEQ}^\top)^\top \mathbf{W}) + \mathrm{tr}((\mathbf{QEQ}^\top)^\top \mathbf{QEQ}^\top). \tag{10}$$

Since only $\mathbf{E}$ is trainable in Equation 10, we can regard the right side of Equation 10 as a function of $\mathbf{E}$, denoted as $f(\mathbf{E})$. Thus, we derive the equivalent form of Equation 8 as:

$$\arg\min_{\mathbf{E}} f(\mathbf{E}) = \arg\min_{\mathbf{E}} (\mathrm{tr}(\mathbf{W}^\top \mathbf{W}) - 2\mathrm{tr}((\mathbf{QEQ}^\top)^\top \mathbf{W}) + \mathrm{tr}((\mathbf{QEQ}^\top)^\top \mathbf{QEQ}^\top)). \tag{11}$$

Since $f(\mathbf{E})$ is a convex and non-negative function, the global minimum is attained by solving $\frac{\partial f(\mathbf{E})}{\partial \mathbf{E}} = 0$. Combined with Equation 11, we obtain the optimal solution of f(W) as:

$$\frac{\partial f(\mathbf{E})}{\partial \mathbf{E}} = -2\mathbf{Q}^\top \mathbf{W} \mathbf{Q} + 2\mathbf{E} = 0. \tag{12}$$

Since $\mathbf{E}$ is a diagonal matrix, we obtain the following equation:

$$\mathbf{E} = \mathrm{diag}(\mathbf{Q}^\top \mathbf{W} \mathbf{Q}), \tag{13}$$

and the proof is complete.

## D EXPERIMENTAL DETAILS ON LARGER-SCALE AND HIERARCHICAL VIT BACKBONES

Table 10 and 11 respectively display the comprehensive results of the comparison conducted in Section 3.3 among ViT-L/16, and Swin-B models.

## E EXPERIMENTAL DETAILS ON ABLATION STUDIES

Table 12, 13, and 14 display the complete results of the ablation studies in Section 4.3

Table 12: This table is extended from Table 5 and shows performance comparisons with different sharing strategies. All experiments are conducted on VTAB-1k benchmark using ViT-B/16 backbone.

| Method \ Dataset | Natural | | | | | | | | Specialized | | | | | Structured | | | | | | | | | Mean Total | Params.(M) |
|---|---|---|---|---|---|---|---|---|---|---|---|---|---|---|---|---|---|---|---|---|---|---|---|---|
| | CIFAR-100 | Caltech101 | DTD | Flowers102 | Pets | SVNH | Sun397 | Mean | Camelyon | EuroSAT | Resisc45 | Retinopathy | Mean | Clevr-Count | Clevr-Dist | DMLab | KITTI-Dist | dSpr-Loc | dSpr-Ori | sNORB-Azim | sNORB-Ele | Mean | | |
| SRSA-SOR | 76.7 | 94.7 | 73.4 | 99.4 | 91.9 | 86.0 | 57.6 | 82.8 | 87.9 | 95.5 | 85.9 | 76.1 | 86.4 | 82.5 | 64.2 | 51.8 | 82.7 | 84.9 | 52.7 | 34.2 | 48.3 | 62.7 | 75.1 | 0.15 |
| w/o intra + inter | 75.9 | 94.0 | 72.4 | 99.4 | 91.5 | 85.0 | 57.3 | 82.2 | 87.2 | 95.1 | 85.5 | 75.5 | 85.8 | 81.4 | 63.6 | 51.4 | 82.3 | 83.7 | 52.0 | 33.5 | 47.1 | 61.9 | 74.4 | 0.15 |
| intra + w/o inter | 75.6 | 94.3 | 72.8 | 99.4 | 91.9 | 85.4 | 57.2 | 82.4 | 87.5 | 95.3 | 85.0 | 75.7 | 85.9 | 82.0 | 63.9 | 51.9 | 82.1 | 83.7 | 52.3 | 34.1 | 47.8 | 62.2 | 74.6 | 0.15 |

Table 13: This table is extended from Table 6 and shows performance comparisons with different adapter positioning. All experiments are conducted on VTAB-1k benchmark using ViT-B/16 backbone.

| Method \ Dataset | Natural | | | | | | | | Specialized | | | | | Structured | | | | | | | | | Mean Total | Params.(M) |
|---|---|---|---|---|---|---|---|---|---|---|---|---|---|---|---|---|---|---|---|---|---|---|---|---|
| | CIFAR-100 | Caltech101 | DTD | Flowers102 | Pets | SVNH | Sun397 | Mean | Camelyon | EuroSAT | Resisc45 | Retinopathy | Mean | Clevr-Count | Clevr-Dist | DMLab | KITTI-Dist | dSpr-Loc | dSpr-Ori | sNORB-Azim | sNORB-Ele | Mean | | |
| LoRA($\mathbf{W}_q, \mathbf{W}_v$) | 67.1 | 91.4 | 69.4 | 98.8 | 90.4 | 85.3 | 54.0 | 79.5 | 84.9 | 95.3 | 84.4 | 73.6 | 84.6 | 82.9 | 69.2 | 49.8 | 78.5 | 75.7 | 47.1 | 31.0 | 44.0 | 59.8 | 72.3 | 0.29 |
| SRSA-SOR($\mathbf{W}_q, \mathbf{W}_v$) | 73.6 | 93.7 | 71.7 | 99.3 | 91.4 | 82.7 | 57.3 | 81.4 | 85.9 | 95.5 | 84.4 | 75.4 | 85.3 | 80.3 | 63.3 | 47.5 | 78.6 | 80.8 | 50.8 | 31.2 | 45.1 | 59.5 | 73.0 | 0.08 |
| SRSA-SOR($\mathbf{W}_q, \mathbf{W}_v, \mathbf{W}_{FC_1}, \mathbf{W}_{FC_2}$) | 74.5 | 94.0 | 71.5 | 99.2 | 91.6 | 86.2 | 57.7 | 82.1 | 87.2 | 96.0 | 86.3 | 75.5 | 86.3 | 81.0 | 63.8 | 48.5 | 80.7 | 83.3 | 52.9 | 31.5 | 45.7 | 61.0 | 74.1 | 0.20 |
| SRSA-SOR($\mathbf{W}_q, \mathbf{W}_k, \mathbf{W}_v, \mathbf{W}_o$) | 76.7 | 94.7 | 73.4 | 99.4 | 91.9 | 86.0 | 57.6 | 82.8 | 87.9 | 95.5 | 85.9 | 76.1 | 86.4 | 82.5 | 64.2 | 51.8 | 82.7 | 84.9 | 52.7 | 34.2 | 48.3 | 62.7 | 75.1 | 0.15 |

Table 14: This table is extended from Figure 3 and shows performance comparisons with different bottleneck dimensions of adaptation matrices in SRSA. All experiments are conducted on VTAB-1k benchmark using ViT-B/16 backbone.

| Method \ Dataset | Natural | | | | | | | | Specialized | | | | | Structured | | | | | | | | | Mean Total | Params.(M) |
|---|---|---|---|---|---|---|---|---|---|---|---|---|---|---|---|---|---|---|---|---|---|---|---|---|
| | CIFAR-100 | Caltech101 | DTD | Flowers102 | Pets | SVNH | Sun397 | Mean | Camelyon | EuroSAT | Resisc45 | Retinopathy | Mean | Clevr-Count | Clevr-Dist | DMLab | KITTI-Dist | dSpr-Loc | dSpr-Ori | sNORB-Azim | sNORB-Ele | Mean | | |
| LoRA($\mathbf{W}_q, \mathbf{W}_v$) | 67.1 | 91.4 | 69.4 | 98.8 | 90.4 | 85.3 | 54.0 | 79.5 | 84.9 | 95.3 | 84.4 | 73.6 | 84.6 | 82.9 | 69.2 | 49.8 | 78.5 | 75.7 | 47.1 | 31.0 | 44.0 | 59.8 | 72.3 | 0.29 |
| Bias | 72.8 | 87.0 | 59.2 | 97.5 | 85.3 | 59.9 | 51.4 | 73.3 | 78.7 | 91.6 | 72.9 | 69.8 | 78.3 | 61.5 | 55.6 | 32.4 | 55.9 | 66.6 | 40.0 | 15.7 | 25.1 | 44.1 | 62.1 | 0.14 |
| D=0 | 72.2 | 90.3 | 70.6 | 99.2 | 91.5 | 58.8 | 57.0 | 77.1 | 81.8 | 93.5 | 81.2 | 74.4 | 82.7 | 56.1 | 45.3 | 40.1 | 73.8 | 56.5 | 44.6 | 22.2 | 30.0 | 46.1 | 65.2 | 0.05 |
| D=1 | 73.6 | 93.7 | 71.7 | 99.3 | 91.4 | 82.7 | 57.3 | 81.4 | 85.9 | 95.5 | 84.4 | 75.4 | 85.3 | 80.3 | 63.3 | 47.5 | 78.6 | 80.8 | 50.8 | 31.2 | 45.1 | 59.5 | 73.0 | 0.08 |
| D=2 | 72.5 | 93.8 | 72.7 | 99.3 | 91.4 | 82.6 | 57.2 | 81.4 | 86.2 | 95.4 | 84.6 | 75.2 | 85.4 | 80.7 | 62.1 | 48.0 | 79.3 | 80.8 | 51.5 | 31.0 | 45.3 | 59.8 | 73.1 | 0.12 |
| D=4 | 73.0 | 93.9 | 72.0 | 99.4 | 91.8 | 83.5 | 57.6 | 81.6 | 86.5 | 95.7 | 84.7 | 75.7 | 85.7 | 81.2 | 63.0 | 48.0 | 80.5 | 80.8 | 50.8 | 31.7 | 45.7 | 60.2 | 73.5 | 0.20 |

