# OpenReview forum: "Enhancing LoRA with Shared Random-Span Augmentation for Parameter-Efficient Visual Adaptation"
_ICLR.cc/2026/Conference — ICLR 2026 Conference Withdrawn Submission_

### Official Review · Reviewer_J7VV · 2025-10-20

**Soundness:** 2
**Presentation:** 3
**Contribution:** 3
**Rating:** 4
**Confidence:** 4

**Summary:**

This paper introduces Shared Random-Span Augmentation (SRSA), a novel parameter-efficient fine-tuning method for vision pre-trained models. SRSA addresses the rigidity of fixed-rank LoRA by replacing learnable unitary matrices with frozen, layer-shared random matrices and learning only diagonal scaling vectors, while retaining a lightweight rank-1 LoRA adapter to capture directional information. This combination preserves LoRA’s low-parameter efficiency while enabling flexible rank adaptation and enhanced representational capacity.

**Strengths:**

1. The paper introduces a simple yet effective enhancement to LoRA by combining fixed, layer-shared random matrices with a lightweight rank-1 LoRA adapter,  achieving flexible rank adaptation.
2. The paper provides theoretical analysis showing that, under a fixed orthogonal basis, learning only a diagonal scaling matrix can optimally approximate full-rank weight updates while preserving their spectral properties, thereby justifying the design of using frozen shared random matrices with learnable scaling in SRSA.

**Weaknesses:**

1. The proposed SRSA method incorporates a rank-1 LoRA adapter, which, according to the ablation studies, plays a crucial role in achieving the reported performance. Although the authors claim that the rank-1 LoRA adapter is designed to capture directional information, the underlying mechanism remains unclear and requires further clarification. Moreover, methods shown in Figure 2, such as VeRA and RandLoRA, could also be combined with a similar rank-1 LoRA adapter, potentially achieving comparable or even better results; thus, additional experiments are needed to validate SRSA’s unique contribution.

2. The experimental baselines are somewhat confusing. The paper does not include direct comparisons between SRSA and similar methods such as VeRA and RandLoRA shown in Figure 2. Even if these approaches were originally not evaluated on visual tasks, the authors should reproduce them under comparable experimental settings for fairness. Furthermore, several important visual fine-tuning methods, including MLAE [1], GLoRA [2], SPT [3], RepAdapter[4], and NoAH [5], are not discussed or compared, even though they report results on VTAB-1k. These works should be properly cited and experimentally compared, and the authors should note that the average accuracy in these papers is calculated differently from that in SRSA when making performance comparisons.

3. The ablation in Table 6 (Section 4.6) is incomplete. Beyond the standard configuration involving q and v, the authors should individually examine the effects of applying SRSA to k, o, fc1, and fc2 to determine which parameters contribute most to performance.

------

[1] MLAE: Masked LoRA Experts for Visual Parameter-Efficient Fine-Tuning. https://arxiv.org/abs/2405.18897

[2] One-for-All: Generalized LoRA for Parameter-Efficient Fine-tuning. https://arxiv.org/abs/2306.07967

[3] Sensitivity-Aware Visual Parameter-Efficient Fine-Tuning. https://arxiv.org/abs/2303.08566

[4] Towards Efficient Visual Adaption via Structural Re-parameterization. https://arxiv.org/abs/2302.08106

[5] Neural Prompt Search. https://arxiv.org/abs/2206.04673

**Questions:**

1. Although I understand that SRSA is proposed primarily for vision pre-trained models, the paper does not provide evidence that the method is limited to visual tasks. Considering that parameter-efficient fine-tuning techniques are more widely applied in large language models, the authors are encouraged to extend SRSA to models such as LLaMA3-8B or Qwen-2.5-7B and report corresponding results.

2. The paper should include a comprehensive comparison of computational efficiency between SRSA and other baselines.

---

### Official Review · Reviewer_7BqD · 2025-10-26

**Soundness:** 3
**Presentation:** 3
**Contribution:** 2
**Rating:** 2
**Confidence:** 4

**Summary:**

This paper proposes a parameter-efficient fine-tuning (peft) method that combines rank-1 LoRA with a new additive term that calculates $W_r \operatorname{diag}(\lambda) W^T_r$ where $W$ is a random (optionally: orthonormal) matrix that is shared between layers and kept frozen during training. This method reduces trainable parameters wrt. LoRA, because only the rank-1 LoRA parameters and $\lambda \in \mathbb{R}^d$ are trainable, i.e., the count of trainable parameters is $3d$ per parameter, whereas rank-$r$ LoRA requires $2rd$ parameters. Experiments on vision tasks (FGVC and VTAB) for a range of pretrained image encoders yield improved performance compared to a number of baselines.

**Strengths:**

**(S1)** The proposed method effectively enables full-rank weight deltas while even reducing the number of trainable parameters compared to LoRA.

**(S2)** The proposed method retains favorable properties of LoRA, such as the possibility to merge weights with pretrained weights for effective deployment

**(S3)** The limitations section is honest and upfront about the limitations of the proposed method. While this does not mitigate the limitations themselves, this transparency should be held in favor of the paper.

**(S4)** The paper is technically sound, and the supplementary material reports a detailed training setup and hyperparameters. The set of baselines is comprehensive and sufficient.

**(S5)** Visuals and explanations in the paper are clear and help in understanding the proposed method.

**Weaknesses:**

**(W1)** The main weakness is the limited experimental evaluation: The proposed method is only evaluated on vision benchmarks. However, it is standard to evaluate on a range of tasks and modalities, including language, especially LLMs. The method clearly is as general as LoRA, so the evaluation should match previous works such as LoRA, DoRA, or VeRA.

**(W2)** The paper does not include an analysis of the additional runtime and memory required by the (frozen) full rank matrices introduced by the proposed method. The computational aspect is important to consider for peft methods.

**(W3)** As also discussed in the paper's limitations section, the method's weight sharing limits reusability between layers of different shapes. The paper offers perspectives on how to mitigate this issue, but it remains a disadvantage compared to methods like LoRA that do not have this constraint.

**(W4)** The benefits of different effective ranks per layer that can be selected dynamically are used as motivation, but there is no analysis of whether the proposed method effectively solves the problem. For example, how will the distribution of column-vector cosine similarities look for SRSA? Does SRSA effectively enable better geometry of the learnt weight delta?

**(W5)** Advantages over baselines are marginal (Tab. 1 and Tab. 2). For FGVC, an additional problem is that the benchmark is approaching saturation, so it becomes hard to observe significant differences between methods. Together with the missing evaluation of computational requirements, this does not allow for concluding with certainty that the proposed method is the best choice among the presented baselines.

**(W6)** Theorem 3.1 is correct, but the proof in Appendix C.1 is not complete. In particular, the transition from Eq. 12 and Eq. 13 as stated does not directly follow. A clearer way is to first show that $\lVert W - QEQ^T\rVert_F^2 = \lVert Q^TWQ - E\rVert_F^2$  and then expand the definition of Frobenius norm as sum of squared entries.  The off-diagonal elements do not affect the minimization, i.e. we only have to find the minimum $\sum_i \left(A_{ii} - E_{ii}\right)^2$ where $A = Q^TWQ$, which directly gives the theorem.

**(W7)** Minor weakness: There are several typos throughout the paper, for example: "ro" (line 307), "dowm" (Fig 2d), and "Structed" (Tab. 3-6, and Fig. 3). Please carefully check the paper for such typos.

**Questions:**

My main suggestion is to significantly expand the evaluation:
  * Include additional tasks and modalities, especially language and LLMs. Other interesting objections are vision tasks beyond classification, such as segmentation or few-shot finetuning of diffusion models. There are many options described in the literature.
  * Discuss and compare the effective computational requirements of the proposed method (memory, speed, FLOPs) with respect to baselines
  * Clarify if the proposed method effectively enables flexible rank allocation in different layers, i.e. if it directly addresses the motivation. This will significantly improve the storyline.

I think addressing these points is necessary to adequately appreciate the strengths of the proposed method.

---

### Official Review · Reviewer_P2Xp · 2025-10-31

**Soundness:** 2
**Presentation:** 2
**Contribution:** 1
**Rating:** 2
**Confidence:** 4

**Summary:**

The paper presents a new PEFT method building upon LoRA. The paper claims to improve LoRA's expressiveness with a small number of additional parameters. The core method focuses on learning an additional diagonal matrix that is multiplied by shared orthonormal matrices. The paper presents results on several vision benchmarks.

**Strengths:**

1. The paper focuses on the relevant problem of improving the expressiveness of LoRA. The paper attempts to achieve this by increasing the rank of the resultant updates. The paper proposes using a high-rank matrix parameterized only by its diagonal values.

**Weaknesses:**

1. Theorem 3.1 is a well-known result in matrix analysis. A more general problem was solved using a similar technique in established works like [1]. The paper should tone down the mathematical claims involving this result.

2. The experimental evaluation presented in the paper is limited. The paper evaluates on small-scale vision benchmarks (often with 1000 samples) using ViTs or their variants. The paper should evaluate on other domains like language, using larger-scale LLMs (at least 7B parameters). The larger models benefit the most from PEFT-based training.

3. The empirical results reported in the paper seem pretty weak. Across Tables 1 & 2, the mean gain in performance over the baselines is only 0.2-0.5% in accuracy. Are these results statistically significant?

4. I'm slightly confused whether the comparison with baselines is far. The proposed method uses a larger number of parameters due to the introduction of the diagonal matrix. In Line 358, the paper mentions "comparable parameter budget". For a fair comparison, the method should have the same number of parameters. Otherwise, the results should be presented as a tradeoff plot between the parameter count and performance.

5. The paper misses out on comparison with a bunch of the latest baselines, including works mentioned in the related works section, like AdaLoRA and DyLoRA. The paper should also compare with [2], which focuses on having high-rank updates by using structured matrices.

[1] A generalized solution of the orthogonal procrustes problem. Schonemann et al. 1966

[2] Structured Unrestricted-Rank Matrices for Parameter-Efficient Fine-tuning. Sehanobish et al, 2024.

**Questions:**

Please respond to the questions in the previous sections.

---

### Official Review · Reviewer_y2vd · 2025-10-31

**Soundness:** 2
**Presentation:** 3
**Contribution:** 1
**Rating:** 2
**Confidence:** 4

**Summary:**

In my understanding, this method enhances LoRA by making rank adaptation lightweight and efficient. First, the LoRA framework is extended with a rank-adaptation mechanism, but instead of learning large unitary matrices, the authors replace them with fixed random matrices. Then they add a sharing strategy so that the same random matrix is reused across layers, reducing memory overhead. Finally, they introduce a small learnable scaling vector to adjust the influence of these random spans, which provides flexibility without heavy computation. This design keeps LoRA’s directional updates while adding adaptability for diverse tasks.

**Strengths:**

The paper has the following strengths:

1. Presentation Quality:
The paper uses well-designed figures and tables that make comparisons and results easy to interpret.


2. Presentation Clarity:
Concepts like LoRA limitations, random-span augmentation, and rank adaptation are explained in a clear and structured way, making the method understandable.

3. Motivation:
The method is well-motivated.

4. Performance within the Experimental Setup:
Experiments across multiple benchmarks (FGVC and VTAB-1k) show consistent improvements over LoRA and other baselines, validating the effectiveness of the proposed approach, given the experimental setup provided by authors.

**Weaknesses:**

W1. Novelty:
The paper does not convincingly differentiate itself from prior LoRA methods. Many works already train diagonal singular value matrices or use adaptive ranks for flexibility, which are not cited.


A. SVD diagonal matrix methods:
The authors don't mention(let alone benchmark against) methods like SVDiff, LoRA-X (which use a learnable diagonal matrix, and Use the SVD decomposition of the model weights). Without these comparisons, it’s unclear if SRSA is better in accuracy or efficiency.

B. Rank Gating/Adaptive Ranking
Apart from  AdaLoRA, DyLoRA, methods such as FouRA, and SoRA (Soft Low-Rank Adaptation) also target the same problem of rank flexibility. There is no comparison against any of these methods.

W2. Experimental Scope
All experiments are limited to vision classification tasks (FGVC and VTAB-1k). The paper does not test on generative tasks (e.g., image generation or diffusion models), which are major LoRA application areas.

Related Work (Please see all related work which mentions these methods, as a lot of papers train on the diagonal space of SVD):
SVDiff: Compact Parameter Space for Diffusion Fine-Tuning
LoRA-X: Bridging Foundation Models with Training-Free Cross-Model Adaptation
FouRA: Fourier Low-Rank Adaptation
SoRA: Sparse Low-rank Adaptation of Pre-trained Language Models

**Questions:**

Why is a random matrix better than matrix decomposition of the base model weights (as shown in SVDiff)?

---

### Note · Authors · 2025-11-12

I have read and agree with the venue's withdrawal policy on behalf of myself and my co-authors.